# Aflatoxin B1 and Aflatoxin M1 Induce Compromised Intestinal Integrity through Clathrin-Mediated Endocytosis

**DOI:** 10.3390/toxins13030184

**Published:** 2021-03-02

**Authors:** Yanan Gao, Xiaoyu Bao, Lu Meng, Huimin Liu, Jiaqi Wang, Nan Zheng

**Affiliations:** 1Key Laboratory of Quality & Safety Control for Milk and Dairy Products of Ministry of Agriculture and Rural Affairs, Institute of Animal Sciences, Chinese Academy of Agricultural Sciences, Beijing 100193, China; gaoyanan0116@126.com (Y.G.); xbao@ualberta.ca (X.B.); menglu@caas.cn (L.M.); liuhuimin02@caas.cn (H.L.); wangjiaqi@caas.cn (J.W.); 2Laboratory of Quality and Safety Risk Assessment for Dairy Products of Ministry of Agriculture and Rural Affairs, Institute of Animal Sciences, Chinese Academy of Agricultural Sciences, Beijing 100193, China; 3Milk and Milk Products Inspection, Center of Ministry of Agriculture and Rural Affairs, Institute of Animal Sciences, Chinese Academy of Agricultural Sciences, Beijing 100193, China; 4State Key Laboratory of Animal Nutrition, Institute of Animal Sciences, Chinese Academy of Agricultural Sciences, Beijing 100193, China

**Keywords:** aflatoxin B1, aflatoxin M1, intestinal epithelial barrier, endocytosis

## Abstract

With the growing diversity and complexity of diet, humans are at risk of simultaneous exposure to aflatoxin B1 (AFB1) and aflatoxin M1 (AFM1), which are well-known contaminants in dairy and other agricultural products worldwide. The intestine represents the first barrier against external contaminants; however, evidence about the combined effect of AFB1 and AFM1 on intestinal integrity is lacking. *In vivo*, the serum biochemical parameters related to intestinal barrier function, ratio of villus height/crypt depth, and distribution pattern of claudin-1 and zonula occluden-1 were significantly affected in mice exposed to 0.3 mg/kg b.w. AFB1 and 3.0 mg/kg b.w. AFM1. *In vitro* results on differentiated Caco-2 cells showed that individual and combined AFB1 (0.5 and 4 μg/mL) and AFM1 (0.5 and 4 μg/mL) decreased cell viability and trans-epithelial electrical resistance values as well as increased paracellular permeability of fluorescein isothiocyanate-dextran in a dose-dependent manner. Furthermore, AFM1 aggravated AFB1-induced compromised intestinal barrier, as demonstrated by the down-regulation of tight junction proteins and their redistribution, particularly internalization. Adding the inhibitor chlorpromazine illustrated that clathrin-mediated endocytosis partially contributed to the compromised intestinal integrity. Synergistic and additive effects were the predominant interactions, suggesting that these toxins are likely to have negative effects on human health.

## 1. Introduction

In high humidity and warm environmental temperatures, crops such as corn, peanut, and wheat are highly likely to encounter air-borne or insect-borne contamination of toxigenic fungi (molds) and their mycotoxins during growth and harvest [1]. There is evidence that over 25% of agricultural products worldwide are contaminated by mycotoxins, and their intake represents the major source of exposure [2]. There is also concern about human mycotoxin exposure by inhalation in certain indoor spaces such as agricultural products processing places and moldy buildings [3,4]. Climate change will significantly influence the amount and diversity of mycotoxins. In the next 25 to 50 years, it is expected that the concentration of CO_2_ in the atmosphere will double or triple, and the temperature will rise by 2 to 5 °C. This will increase the frequency of droughts and promote plant stress, which is likely to affect the production of secondary metabolites, especially mycotoxins [5].

Most of the existing toxicological data are based on individual toxic substances, but humans are exposed to multiple chemical contaminants in the environment [6]. Among these chemical contaminants in the environment, mycotoxins are omnipresent human health hazards. It has been expounded that the co-occurrence of mycotoxins in the environment is a rule rather than an exception [7]. In this scenario, it is important to measure the toxicities of combined mycotoxins, which could exert additive, synergistic, or antagonistic interactive effects [8].

Aflatoxin B1 (AFB1), produced by *Aspergillus parasiticus* and *Aspergillus flavus*, is found worldwide in agricultural products. Due to its potent carcinogenicity, the International Agency for Research on Cancer (IARC) classified AFB1 as a Group 1 carcinogen to humans [9]. After ingestion, more than 80% of the absorption of AFB1 occurs in the proximal gastrointestinal tract in the vein by passive transportation [10]. It is then delivered to organs throughout the body, causing damage and dysfunction. The metabolism and activation of AFB1 can take place in the liver by cytochrome P450 (CYP) enzymes (mainly by CYP1A2 and CYP3A4) [11]. While it has been largely described in the liver, AFB1 metabolism to its toxic metabolite also occurs in the digestive tract [12]. Apart from the epoxidation, it can also be oxidized by P450-dependent monooxygenases to hydroxylated products, such as aflatoxin M1 (AFM1), which is the most threatening component of AFB1 contamination [13]. AFM1 could be present in the milk of dairy cows and human nursing mothers and in mammals that consume a diet containing AFB1. IARC has modified the carcinogen classification of AFM1 from Group 2B to Group 1 [9]. High AFM1 occurrence and contamination levels in milk samples were reported [14,15,16]. With the growing diversity and complexity of the human diet, humans could be exposed to AFB1 and AFM1 through daily consumption of contaminated agricultural products, milk, and dairy products. Consequently, there is a risk of co-exposure to both toxins.

The intestinal epithelium with perfect structural integrity serves many crucial functions, guaranteeing the welfare and health of animals and humans. It functions as a filter facilitating the uptake of nutrients from the intestinal lumen [17]. As the first target of external contaminants, it also protects against the entry of various toxic compounds. Intestinal barrier disruption may contribute to the activation of an inflammatory response, thereby threatening human health [18]. Tight junction (TJ) proteins are the main functional components of the intestinal barrier, sealing the intercellular spaces of adjacent epithelial cells [19]. It has been demonstrated that TJ proteins can be internalized through endocytosis, including clathrin- and caveolae-mediated endocytosis and micropinocytosis [20], and subsequently sorted to early recycling endosomes to be recycled back to the plasma membrane, or late endosomes to be degraded [21].

The objectives of the present study were to evaluate the combined toxicity of AFB1 and AFM1 on the intestinal barrier and on mechanisms involved in dysfunction in both *in vivo* and *in vitro* models. *In vivo*, individual and combined effects of AFB1 and AFM1 were assessed in Institute of Cancer Research (ICR) mice ileum through serum biochemical parameters, intestinal histomorphology, and the distribution of TJ proteins. *In vitro*, effects of the toxins on phenotype and mechanism of intestinal barrier function were analyzed in differentiated Caco-2 cells. The *in vivo* and *in vitro* data in the present study demonstrated for the first time that AFB1 and AFM1 led to a compromised intestinal barrier related to clathrin-mediated endocytosis. In addition, our results showed that the main interactive effect between AFB1 and AFM1 on intestinal barrier dysfunction was synergism.

## 2. Results

### 2.1. In Vivo Effects of AFB1 and AFM1 on the Intestinal Barrier

Compared with the control group, AFB1 and AFM1 alone or in combination did not significantly (*p* > 0.05) affect the mice body weight; however, a decrease of about 10% for the combined 28-day treatment was observed (Figure 1). From the results of the serum biochemical test, we can see that the combination of AFB1 and AFM1 significantly (*p* < 0.05) decreased citrulline (Cit) concentrations and increased diamine oxidase (DAO) activity (Figure 2A,B). In addition, the concentrations of intestinal fatty acid-binding protein (I-FABP) and D-lactate were significantly (*p* < 0.05) increased in the combined toxins treatment, suggesting compromised intestinal integrity (Figure 2C,D). Histomorphometrical changes caused by AFB1 and AFM1 were assessed by measuring villus height and crypt depth in the ileum, which could also reflect the intestinal cell proliferation (Figure 3A–D). A significant (*p* < 0.05) reduction in villus height was observed in the ileum of mice exposed to both individual and combined AFB1 and AFM1, whereas the crypt depth was enhanced in the individual AFM1 and combined AFB1+AFM1 treatment (Figure 3E). The ratio of villus height/crypt depth was significantly (*p* < 0.05) depressed in the mice exposed to AFB1 and AFM1 alone and in combination (Figure 3F). To study the effects of AFB1 and AFM1 alone and in combination on the TJ structure of mice intestinal epithelial cells, the abundance of claudin-1 and ZO-1 were assessed in the ileum by immunohistochemical staining. From the results, we observed that a weak immunostaining intensity and a clear collapse of the grid-like barrier structure were shown in the mice exposed to the mixture (Figure 4 and Figure 5).

### 2.2. Cytotoxic Effects of AFB1 and AFM1 on Differentiated Caco-2 Cells

Given that AFB1 and AFM1 affected the indicators of intestinal barrier integrity *in vivo*, experiments were also performed on differentiated Caco-2 cells. As shown in Figure 6A, both toxins provoked a decrease in cell viability in a dose-dependent manner. AFB1 was apparently more potent than AFM1, as AFB1 caused a significant decrease of 10% in cell viability at 1 μg/mL, whereas AFM1 treatment led to a similar decrease in cell viability at 4 μg/mL (Figure 6A). As depicted in Figure 6B, upon exposure to a combination of both toxins for 48 h, synergistic effects (CI: 0.167–0.72) prevailed at lower test concentrations while antagonistic effects (CI: 1.38–29.53) were observed at higher concentrations, indicating that the type of interactive effects varies with toxin concentrations.

### 2.3. Intestinal Permeability Effects of AFB1 and AFM1 on Differentiated Caco-2 Cells

After 48 h of exposure to different concentrations of toxins, a concentration-dependent decrease in trans-epithelial electrical resistance (TEER) values was observed (Figure 7A). Compared with the control group, AFB1 at 2 μg/mL significantly reduced (*p* < 0.05) TEER values, whereas AFM1 at concentrations below 8 μg/mL did not cause a significant drop of TEER values (*p* > 0.05).

To investigate the effects on intestinal permeability induced by individual and combined AFB1 and AFM1 treatments, the paracellular flux of different molecular weights (4 and 40 kDa) fluorescent tracer fluorescein isothiocyanate (FITC)-dextran was measured. The individual toxin AFB1 at 0.5–1 μg/mL and AFM1 at 8 μg/mL were found to exert negative effects on paracellular resistance (Figure 7B,C). Based on the data of cell viability and epithelial integrity (TEER and FITC-dextran), AFM1 (0.5 and 4 μg/mL) and corresponding concentrations of AFB1were selected in the subsequent experiments.

### 2.4. Effects of AFM1 on AFB1-Induced TJs Destruction on Differentiated Caco-2 Cells

As shown in Figure 8A, compared with the control group, AFB1 and AFM1 at low concentrations (0.5 μg/mL) did not significantly (*p* > 0.05) decrease the transcript levels of TJs even when the toxins were administrated as mixed additions. The 4 μg/mL AFB1 caused a significant (*p* < 0.05) reduction in TJ mRNA expression, whereas no such effect was found in 4 μg/mL AFM1.

No significant (*p* > 0.05) inhibition of TJ protein expression was found upon exposure to individual AFB1 and AFM1 at 0.5 μg/mL by western blot analysis (Figure 8B,C). The combination of AFB1 and AFM1 at 4 μg/mL generated significant (*p* < 0.05) destruction of TJ proteins, whereas ZO-1 seemed to be more vulnerable, as a significant reduction of its abundance was induced by 0.5 μg/mL mixed toxins and individual toxins at 4 μg/mL (*p* < 0.05). Compared with the control group, the mixture of AFB1 and AFM1 at higher concentrations produced more serious damage of TJ proteins than individual toxins.

The localization and cellular distribution of TJ proteins were determined by immunofluorescent staining to confirm the results of immunoblotting. In the control group, TJ proteins were located in the cell–cell junctions and formed an anastomosing network (Figure 8D). Following the treatment with AFB1 and AFM1, all four TJ proteins examined disappeared from the plasma membrane as evidenced by a faint immunofluorescent signal and weak network, suggesting that the synthesis and/or redistribution processes of these proteins were affected.

### 2.5. Involvement of Clathrin-Mediated Endocytosis in the Redistribution of TJ Proteins Induced by AFB1 and AFM1

Given that the changes of TJ protein localization may be associated with the change of plasma membrane integrity, lactate dehydrogenase (LDH) release from differentiated Caco-2 cells into the culture medium was determined to assess whether exposure to toxins caused plasma membrane damage. Results showed that treatment of cells with 4 μg/mL AFB1 or combined with 4 μg/mL AFM1 produced significant elevations of LDH leakage compared with the control group (Figure 9A).

To explore the mechanism by which the toxins (individual or combined) compromised intestinal integrity, inhibition of clathrin-mediated endocytosis chlorpromazine (CP) was carried out following AFB1 and AFM1 exposure. Figure 9B showed that compared with the combined mycotoxins treatment, the immunofluorescence intensity was stronger in the presence of preincubation with CP, especially for claudin-4. In addition, as shown in Figure 9C, TEER values were not significantly changed after treatment with 60 μM CP alone. AFB1-induced TEER decrease was significantly ameliorated by CP pretreatment, suggesting that clathrin-mediated endocytosis was involved in AFB1-induced damaged intestinal barrier integrity. A tendency in improved TEER values was also observed with the pretreatment of CP followed by exposure of the mixed toxins for 48 h. This finding implies that clathrin-mediated endocytosis partly resulted in compromised intestinal integrity induced by combined AFB1 and AFM1 treatment.

### 2.6. Correlations between Intestinal Permeability and TJ Proteins

To verify the significance of changes in intestinal permeability and TJ protein levels, a correlation analysis among TEER values, FITC-dextran (4 and 40 kDa), and expression of TJ proteins (occludin, claudin-3, claudin-4, ZO-1) was conducted. As shown in Figure 10, a significant (*p* < 0.05) positive correlation was found in TEER values and TJ proteins, indicating that compromised intestinal barrier integrity is related to the disruption of TJ proteins.

### 2.7. Interactive Effects in the Combination of AFB1 and AFM1

The interactive effects of AFB1 and AFM1 at different concentrations were assessed according to common rules for interactive analysis mentioned in Section 4.13. We found an additive effect in TEER value, claudin-3, claudin-4, and ZO-1 with nonsignificant differences (*p* > 0.05) between the expected and measured values (Figure 11A,D,E,G). As for the paracellular flux of FITC-dextran (4 and 40 kDa), an additive effect between 0.5 μg/mL AFB1 and AFM1, whereas a synergistic effect was determined after treatment with AFB1 and AFM1 at 4 μg/mL collectively. The expected values were 98.3% and 90.3% (*p* < 0.05) lower compared to measured values, respectively (Figure 11B,C). We observed an antagonistic effect on occludin expression with combined AFB1 and AFM1 treatment, as the measured values were 19.0% and 18.9% (*p* < 0.05) higher than the expected values, respectively (Figure 11F).

## 3. Discussion

With the increasing daily consumption of milk and the diversity of the human diet, the potential risk of exposure to the toxins AFB1 and AFM1 is of great concern. Due to its particular location and function, the intestinal epithelium is regarded as the main target of potential negative effects induced by mycotoxins [19]. Therefore, it is important to define the underlying mechanism(s) of the effects of AFB1 and AFM1 on intestinal integrity as well as evaluate their interactive effects. *In vivo* (ICR mice) and *in vitro* (differentiated Caco-2 cells) methods were used in the present study. We found that AFB1 and AFM1 induced intestinal integrity dysfunction in differentiated Caco-2 cells through clathrin-mediated endocytosis on TJ proteins. The main interaction types between AFB1 and AFM1 on the intestinal barrier were synergistic and additive effects.

During the past few decades, acute or short-term liver and renal toxicity have been studied in recent rodent studies with single or short-term AF (mainly AFB1) oral dosing (AFB1, 0.5–5 mg/kg b.w.) [22,23,24,25,26,27]. Recently, chronic toxicity studies on animals (at lower dose levels) revealed carcinogenic effects of AFB1 and/or AFM1 on the liver, lung, and gastrointestinal tract (reviewed in [28]). Whilst it is clearly of interest to further examine lesions induced by these toxins in multiple organs, it is worth noting that present toxicological risk assessments for AFs are based on the mode of action considerations and the most critical effect found at very low chronic doses, i.e., the induction of liver cancer [29,30]. A previous study demonstrated that AFB1 residues in the blood of the AFB1-fed treatment were significantly higher (*p* < 0.05) than those fed with standard mouse feed for 14 days [31]. Similar results also showed in the Juvenile Turbot fed with AFB1 diets [32]. Not only in the blood, mycotoxins could also cause residue in various tissues in mice and rats, such as liver and kidney [33,34,35].

A critical reflection about the doses chosen in the *In vivo* part of the present study and concentrations applied to Caco-2 cells is indicated: these are several orders of magnitude higher than toxin levels found in the serum of human beings (reviewed in [36]). The levels of AFB1-lysine (AFB1-lys) and AFB1-albumin adducts are also regarded as the biomarker of AFB1 in human plasma and serum samples. It has been reported that a mean concentration of AFB1-lys adducts was 20.5 ± 23.4 pg/mg in 67% of Malawian rural population blood samples (*n* = 230). According to regression analysis, the consumption of groundnut (AFB1:52.4 μg/kg) contributes to elevated AFB1-lys adducts, with a mean concentration of 20.5 ± 23.4 pg/mg of albumin [37]. AFB1-lys and AFM1 were not found in the participants’ serum samples in Brazil, whereas AFM1 was detected in 65% of urine samples. The 232 food products (peanut products and milk products) provided by these participants were contaminated by AFB1 and AFM1 with concentrations under the tolerance limits established by Brazilian regulations [38]. Not only food-borne exposure but also the contaminated environment may lead to the intake of toxins for humans. There was a significant correlation between dust and AFB1 concentrations. The concentrations of AB1-albumin in the serum of workers in plastic and bread waste-storing sections were 3.83 ± 0.49 and 4.02 ± 0.63 μg/mg albumin, respectively [39]. These findings suggest that the intake of AFs may be inevitable. In such scenarios, in addition to the well-known liver cancer risk from low-level chronic exposure to AFs, other adverse effects on human health are anticipated [40,41].

Although the body weight of mice exposed to individual toxins was not significantly (*p* > 0.05) affected, a decrease of about 10% in the combined AFB1 and AFM1 treatment on body weight was observed, indicating some toxicity (Figure 1). Moreover, alterations in serum biochemical parameters related to intestinal barrier function, morphology, and TJ distribution of the ileum were observed, indicating that AFB1 and AFM1 induced compromised intestinal barrier (Figure 2, Figure 3, Figure 4 and Figure 5). In certain clinical conditions, the plasma content of Cit, DAO, D-lactate, and I-FABP have been regarded as reliable markers of intestinal integrity [42,43,44,45]. However, these parameters have not been widely used to evaluate AF-induced effects, with only two related studies reported. Serum levels of D-lactate in broiler chicks with a 1.5 mgAFB1/kg diet for 20 days were significantly enhanced [46]. Serum DAO concentrations were increased in broilers fed with 40 μg/kg AFB1 contaminated feed [47].

Adverse effects of AFs, especially for AFB1, on intestinal morphology, including decreased villus height or increased crypt depth, have been recorded in various *in vivo* models. Wistar rats intraperitoneally administered with 2.5 mg/kg AFB1 for 7 days experienced villi degeneration of the duodenum and ileum [48]. Increased crypt depth of the duodenum was observed in the ducks fed with 195.4 μg/kg AFB1 contaminated feed [49]. Considerable decreases of the jejunal villi height to crypt depth ratio in broiler chicks exposed to 0.5 and 2 mg AFs/kg feed for 28 or 42 days were reported [50]. However, villus length in both duodenum and ileum sections was significantly increased in a dose-related fashion in the broiler chickens that received 1 and 1.5 ppm of AFB1 contaminated feed for 21 days [51]. Increased villus height of jejunum was also observed in the ducks fed with 195.4 μg/kg AFB1 contaminated feed [49]. These contradictory results indicate that the intestinal morphology varies between the species of animals used, the section of intestine chosen, and the concentration of toxins. It would also be interesting to compare the response of different intestinal tissue sections to toxins. Considering that an intestinal disruption could also result in inflammatory responses, it is worth measuring serum cytokines as markers of inflammation in our future studies.

In this study, we demonstrated the toxicity of individual toxins on differentiated Caco-2 cell viability in a concentration-dependent manner, and AFM1 seemed less potent than AFB1. This may be related to the fact that AFB1 is shown to be the more active compound, which is more potent than AFM1 with respect to liver carcinogenicity by approximately 10-fold [30]. In addition, this result is in accordance with previous studies [52], although there are some differences in the concentrations of AFM1 and AFB1 at which a significant decrease in cell viability starts to be recorded. In the present study, AFB1 and AFM1 led to cytotoxicity at 1 μg/mL and 4 μg/mL, respectively. Another study demonstrated that 0.0005–4 μg/mL AFM1 had no effect on the viability of differentiated Caco-2 cells for 48 h [53]. By using the Alamar Blue assay, the viability of Caco-2 cells decreased with increasing doses of AFB1 from 10 to 50 μM (correspond to 3 to 15 μg/mL) for 24 h [54]. AFM1 caused a clear decrease in Caco-2 cell viability between 3 and 12 μM, corresponding to 1 to 4 μg/mL [55]. The variable outcomes could be partially explained by (i) different properties of differentiated and undifferentiated Caco-2 cells, which might be due to the alteration of enzymes during cell differentiation in 16–22 days, (ii) exposure time, and (iii) detection method. On the other hand, AFM1 at 10 to 10,000 ng/kg medium (about 0.3 to 32 nM) did not cause decreases in the viability of differentiated and undifferentiated Caco-2 cells [56]. This is no surprise because these concentrations were far below (about 1000-fold) those used in the present study.

TEER values and paracellular flux of FITC-dextran (4 and 40 kDa) are generally regarded as indicators of intestinal epithelial integrity [57]. In the present study, AFB1 and AFM1 increased differentiated Caco-2 cell permeability at 2 µg/mL and 8 µg/mL, respectively. TEER values significantly decreased in differentiated Caco-2 cells following incubation with 150 µM (45 µg/mL) AFB1 for 72 h [58]. After treatment of differentiated Caco-2 cells with AFB1 (100 µM) for 7 days, TEER values decreased by up to 51% [59]. Caloni et al. [60] demonstrated that exposure to AFM1 (from 1000 to 10,000 ng/kg) for 6 h in the differentiated Caco-2 cell monolayer led to a slight but significant (*p* < 0.05) TEER decrease, not in a dose-dependent way, and the TEER values were basically unchanged after 48 h.

TJ proteins, the multiple proteins, include transmembrane proteins such as claudins, occludin, and intracellular plaque proteins such as ZO-1 [61]. In the present study, 0.5 μg/mL AFM1 tends to increase the transcription expression of claudin-3. This is consistent with a study that showed that an increase in transcript levels of claudin-1 and claudin-2 in the jejunum of broiler chicks fed with 1.5 mg/kg AFB1 for 20 days was observed [46], which might be related to continuous rather than acute exposure to toxins. A significant (*p* < 0.05) decrease in the transcription levels and protein abundance of TJs was observed after exposure to AFB1 and AFM1 at 4 μg/mL. A previous study has reported that AFB1 at 3–30 μM decreased transcript level of claudin-3 and occludin in differentiated Caco-2 cells [59], consistent with the present study results. In our study, confocal microscopy showed that the location of TJ proteins became disordered and scattered compared to the original morphological cobblestone structure, consistent with previous studies demonstrating changes in intestinal epithelial cell permeability associated with protein abundance and subcellular localization [62,63]. The Spearman’s correlation assessment conducted indicates significant correlations between TEER values, FITC paracellular flux, and four examined TJ proteins. We concluded that the impaired barrier integrity function of differentiated Caco-2 cell monolayers by AFB1 and AFM1 exposure was closely related to the alteration of the expression of TJs and their localization.

From the above results of TEER values, FITC-dextran, and TJ proteins, we confirmed that AFB1 and AFM1 compromised intestinal barrier function; however, the exact underlying mechanism is still unknown. In addition to the above-mentioned deregulated expression of TJs and the changed localization between cells, TJs were observed internalized to some extent into the cytoplasm. The internalization of TJ proteins has been regarded as a pivotal mechanism that regulates the plasticity and function of TJs in the epithelial barrier, with endocytosis involved in the TJs internalization process [21]. TJ proteins can be internalized by endocytosis when different epithelial cells are treated with different substances [63,64,65]. There are two types of endocytosis: clathrin independent and clathrin dependent. Among them, clathrin-dependent endocytosis is essential in epithelial cells to remove transmembrane TJ proteins from the plasma membrane, dissociating the complex of TJ proteins [21]. In this study, we observed LDH leakage and clathrin-mediated endocytosis using CP on differentiated Caco-2 cells after exposure to AFB1 and AFM1. Based on these results, we speculate that (i) the cell membrane was damaged and fragmented after toxin treatment, leading to the location of TJ proteins becoming cytoplasmic when observed under confocal microscopy; (ii) clathrin-mediated endocytosis and TJ transport into the cytoplasm might be stimulated by AFB1 and are partially responsible for the aggravation by AFM1 of AFB1-induced intestinal epithelial integrity on differentiated Caco-2 cells. Evidence for TJ protein internalization in Caco-2 cells after exposure to AFB1 and AFM1 was provided for the first time, although the detailed mechanism(s) needed to be explored in depth.

One of the aims of the current study was also to produce a relatively more precise assessment of the interactions among mycotoxins. In our study, we conducted an isobologram analysis based on mass-action quantification [66] as well as a comparison between expected and measured values [67]. The isobologram analysis showed that AFB1 and AFM1 combination at low concentrations exerted synergetic effects, whereas antagonistic effects were observed at high concentrations. AFB1 and AFM1 have similar chemical properties and can cause the same kinds of toxicity (carcinogenicity, cytotoxicity, mutagenicity) but with different potencies. This is possibly related to some quantitative differences in their bioactivation and detoxication [28,68]. Structural similarity is also embraced in our hypothesis that both toxins may compete with the same sensors and receptors. This may explain why antagonistic effects were seen when AFB1 and AFM1 were mixed at high concentrations. According to the comparison between expected and measured values, the predominant interactive effects for the combination of AFB1 and AFM1 on intestinal permeability and TJ proteins expression were synergistic and additive. This indicates that negative effects on intestinal barrier function induced by combined toxins cannot be ignored, which provides new insight on realistic risk assessment of the effects of toxins on human health.

The potential effects of mycotoxins other than AFs on intestinal integrity in *in vivo* and *in vitro* models have been well summarized in recent reviews [19,57]. In addition to combined mycotoxins, humans could also be simultaneously exposed to mycotoxins and heavy metals through food consumption. Reduced jejunal crypt depth and occludin expression were observed in rats exposed to a mixture of deoxynivalenol (DON, 8.2 mg/kg feed) and cadmium (Cd, 5 mg/L water) [69]. This combination exerted additive and antagonistic effects in decreasing Caco-2 cell viability [70]. Further studies could be devoted to the effects of these frequently co-occurring chemical contaminants in the human food chain, which could more closely reflect the potential risk for key human intestinal functions.

## 4. Materials and Methods

### 4.1. Chemicals

AFB1 and AFM1 standard were obtained from J&K (Shanghai, China). They were dissolved in methanol to a compound stock solution of 1 mg/mL and stored at −20 °C. The working dilutions were prepared in a serum-free medium for use, and the final methanol concentration was below 1% (*v*/*v*). A final concentration of 0.8% methanol corresponding to the highest concentration of working dilutions was tested, and there was no significant difference from the control. Tween-20 was obtained from Sigma-Aldrich (St. Louis, MO, USA). Primary and secondary antibody dilution buffer, blocking buffer, and antifade mounting medium were obtained from Beyotime Biotechnology (Shanghai, China). Rabbit anti-occludin, rabbit anti-claudin-1, rabbit anti-claudin-3, rabbit anti-claudin-4, rabbit anti-zonula occludens-1 (ZO-1), rabbit anti-β-actin, and mouse anti-rabbit IgG conjugated to horseradish peroxidase were obtained from Bioss (Beijing, China).

### 4.2. Animals

Forty four-week-old male ICR (CD-1) mice (20 ± 2 g) were purchased from Beijing Vital River Laboratory Animal Technology Co., Ltd. (Beijing, China). The mice were kept under conditions of 21–23 °C, 45–55% humidity with a 12-h day/night cycle. The doses of toxins were chosen based on the half-lethal dose (LD50 values), which was estimated through a preliminary acute oral toxicity experiment based on the standard of GB/T 21826-2008 [71]. The LD50 values for AFB1 were approximately 6 mg/kg b.w., and the dose used in the present study was 1/20 of the LD50 values. AFM1 was proved to induce liver cancer with a potency of one-tenth that of AFB1. In general, for AFM1, a potency factor of 0.1 relative to AFB1 was used [30], so we estimated the LD50 values for AFM1 were 60 mg/kg b.w. Therefore, a dose of 3.0 mg/kg b.w. AFM1 (1/20 of the LD50 values) was chosen in this study. The doses used in the present study are several orders of magnitude higher than toxin levels to the human possible chronic exposure intake. After seven days of acclimatization, forty mice were randomly divided into four groups: a control group (1% DMSO), an individual AFB1 group (0.3 mg/kg b.w.), an individual AFM1 group (3.0 mg/kg b.w.), and a combination group (0.3 mg/kg b.w. AFB1+3.0 mg/kg b.w.). The mice in the treatment groups were gavaged once per day (0.2 mL/mouse) for four weeks and weighed twice a week. At day 29, the mice were euthanized by CO_2_. The intestinal tissue was collected and fixed in 10% buffer formalin for subsequent histological assessment and immunohistochemical staining. Blood samples were gathered from the retro-orbital plexus. All animal procedures were performed according to the Chinese guidelines for animal care, conforming to internationally accepted principles for the care and use of experimental animals. Animal experiments were approved by the Ethics Committee of the Institute of Animal Sciences of Chinese Academy of Agricultural Sciences (Beijing, China, IAS 2019-3, 18 March 2019).

### 4.3. Cell Culture and Treatment

Caco-2 cell lines (passages between 15 and 35) derived from human colon adenocarcinoma were purchased from the American Type Culture Collection (ATCC) (Manassas, VA, USA). They were cultured in Dulbecco’s modified Eagle’s medium (DMEM) with a supplement of 4.5 g/L glucose, antibiotics (100 units/mL penicillin and 100 mg/mL streptomycin), 1% nonessential amino acids (NEAA), and 10% fetal bovine serum (FBS) (Life Technologies, Carlsbad, CA, USA) in a humidified incubator at 37 °C and 5% CO_2_ atmosphere. In order to promote differentiation, Caco-2 cells were maintained on permeable 12-well transwell chambers (Corning, New York City, NY, USA). After 21 days of culture, individual and combined AFB1 and AFM1 were added to the apical compartments of transwell chambers for 48 h.

### 4.4. Serum Biochemical Indicators Determination

The collected blood samples were coagulated at 37 °C for 1–2 h (without anticoagulant) and then placed in a centrifuge at 4 °C, 4000 rpm for 10 min, to obtain the supernatant (serum). The serum biochemical parameters related to intestinal barrier function, including Cit, DAO, I-FABP, and D-lactate, were measured by automatic biochemical analyzer HITACHI 7080 (HITACHI, Tokyo, Japan) and enzyme-linked immunosorbent assay (ELISA).

### 4.5. Histological and Morphometric Assessment of the Ileum in Mice

Samples of ileum were rinsed with normal saline solution to remove intestinal contents and traces of blood and then transversally cut. Thereafter, ileum pieces were fixed in 10% buffered formalin. After 24 h, all the fixed samples were dehydrated in an ascending alcohol series (75–100%). These samples were initially sectioned at a thickness of 4 mm, then embedded in paraffin, followed by cutting 5 μm sections with a rotary microtome. The sections were stained with hematoxylin and eosin (HE) for histopathological evaluation. The ileum tissues of 6 mice in each group were selected, and 3 different positions were taken to collect slices in each section. The morphology of the ileum after HE staining was observed under the microscope (Leica DM500, Leica, Wetzlar, Germany). Villus height and crypt depth were measured in 5 villi from each slice, and the ratio of villus height/crypt depth was also calculated.

### 4.6. Cytotoxicity Assay

The cytotoxicity of the compounds was analyzed using the CCK-8 assay (Beyotime, Shanghai, China) according to the manufacturer’s instructions. In brief, individual and combined AFB1 and AFM1 (0.5, 1, 2, 4, and 8 μg/mL) were added to the apical and basal compartments of transwell chambers. After 48 h, 120 μL CCK-8 operating reagent was added to the apical side of every well in permeable Millicell polycarbonate (PCF) filters. After about 1 h incubation at 37 °C, we pipetted out 100 μL liquor to measure the absorbance at a test wavelength of 450 nm using an automated ELISA reader (Thermo Scientific, Waltham, MA, USA). In the present study, the cell viability following toxin treatment was presented with respect to the control as the following formula:

Viability (%) = 100 × Mean OD in treatment group/Mean OD in control group

### 4.7. TEER Measurement

Initial trans-epithelial electrical resistance (TEER) values were assessed before the addition of toxins using a voltmeter (Millicell-ERS; Millipore, Bedford, MA, USA) and measured again after 48 h incubation. TEER was reported as a percentage of the initial value, set as 100%. The baseline TEER values of the Caco-2 cells after 21-day culture ranged between 1865 and 2176 Ω·cm^2^.

### 4.8. Permeability Measurement

Membrane-impermeable tracers fluorescein isothiocyanate (FITC)-dextran standard of 4 kDa and 40 kDa were dissolved in phosphate-buffered saline (PBS). The final concentrations of FITC-dextran used in this study were 100 µg/mL. The tracers were added to the apical side of the epithelial monolayer for 4 h, and samples were collected from the basolateral chamber and taken to a microplate reader (Thermo Labsystems, Waltham, MA, USA) to determine the intensity of fluorescence. The emission and excitation wavelengths were 520 and 490 nm, respectively. Based on the results of cell viability and epithelial integrity (TEER and FITC-dextran), AFM1 (0.5 and 4 μg/mL) and corresponding concentrations of AFB1 were selected in the subsequent experiments to determine the effects of non-cytotoxic AFM1 on AFB1.

### 4.9. Immunofluorescence Analysis

For immunofluorescence analysis with confocal microscopy, after dewaxing and heat-induced antigen retrieval, the sections of mice ileum were incubated with an anti-claudin-1 and anti-ZO-1 antibody overnight at 4 °C, followed by the secondary antibody. Caco-2 cells (2 × 10^5^ per well) were cultured on coverslips in 6 well plates for 21 days [72,73] and treated with individual and combined AFB1 and AFM1 for 48 h. After being washed with PBS 3 times, the cells were fixed at 4 °C with 4% paraformaldehyde for 10 min and subsequently permeabilized for 10 min, and then washed 3 times with tris buffered saline tween (TBST). After that, they were incubated with blocking buffer (TBST) for 3 h and then washed with TBST 3 times. The cells were incubated with an anti-occludin, anti-claudin-3, anti-claudin-4, and anti-ZO-1 antibody for 2 h at 4 °C on a slow shaker. After shaking, the cells were washed by TBST 3 times, then incubated with Alexa Fluor 488 mouse anti-rabbit IgG for 1 h in the dark. 4′,6-diamidin-2-phenylindol (DAPI) was used to counterstain the nuclei. Fluorescence images were obtained by LSM780 immunofluorescence microscope (Carl Zeiss, Inc., Thornwood, NY, USA).

### 4.10. mRNA Expression Level of the Tight Junction Target Genes

Quantitative real-time PCR (qRT-PCR) was performed to measure the mRNA level of the tight junction target genes in differentiated Caco-2 cells exposed to AFB1 and AFM1 alone or in combination. Total RNA extraction, cDNA conversion, and qRT-PCR analysis were conducted as previously described [59]. The primers for occludin (*OCLN*), claudin3 (*CLDN3*), claudin4 (*CLDN4*), and ZO-1 (*TJP1*) were designed by software Premier 5.0 (PREMIER Biosoft International, Palo Alto, CA, USA) as shown in Table 1 below. Relative mRNA expression was normalized to *GAPDH*.

### 4.11. Western Blot Analysis of the Tight Junction Proteins

Differentiated Caco-2 cells were washed in sterile PBS and lysed in radioimmunoprecipitation (RIPA) buffer for 30 min after 48 h individual and combined toxin treatment. A total of 100 μg protein was separated by SDS-PAGE and transferred to the polyvinylidene fluoride (PVDF) membrane, then blocked with 5% skim milk powder for 1.5 h at room temperature. The membranes were incubated with primary antibodies of 4 different TJ proteins (as mentioned above in Section 4.1) overnight at 4 °C. After that, the membranes were washed and then incubated with appropriate secondary antibodies for 2 h at room temperature. The Image J 2× software (Version 2.1.0, National Institutes of Health, Bethesda, MD, USA, 2006) was used to analyze the band densities. The results of western blot are expressed as mean ± SD of three independent experiments.

### 4.12. Comparison between Measured and Expected Values

As we previously described [62], the expected values were obtained by the following formula:mean (expected for AFB1+AFM1) = mean (AFB1) + mean (AFM1) − 100%
% difference = absolute value of (mean (expected for AFB1+AFM1) − mean (measured for AFB1+AFM1))
SEM (expected for AFB1+AFM1) = [(SEM for AFB1)^2^ + (SEM for AFM1)^2^]^1/2^

### 4.13. Analysis for Interactions and Correlations

The interactive effects of toxins on cell viability were analyzed by isobologram analysis, which allows us to calculate a combination index (CI) value. CI > 1, CI = 1 and CI < 1 indicate antagonistic, additive, and synergistic effect, respectively [66].

The expected values of TEER, the paracellular flux of FITC, and the expression of 4 TJ proteins were calculated and compared to their measured values, respectively, as described above. A synergistic effect was defined when measured values were significantly lower than the expected values for endpoints TEER, occludin, claudin-3, claudin-4, and ZO-1 and significantly higher than for endpoints FITC. An additive effect was defined when there was no significant difference between the expected and measured values (*p* > 0.05). An antagonistic effect was defined in opposition to a synergistic effect.

Correlations among these 7 parameters were analyzed by Spearman’s correlations (nonparametric).

### 4.14. Lactate Dehydrogenase (LDH) Assay

After 48 h exposure, the supernatant samples were collected, and the LDH activity was determined at 490 nm absorbance wavelength by LDH cytotoxicity assay kit (Beyotime, Shanghai, China) as per the manufacturer’s instructions. Total LDH activity was defined as the sum of intracellular and extracellular LDH activity and was normalized as 100%; the amount of LDH in the culture medium was expressed as a ratio of this total value.

### 4.15. Clathrin-Mediated Inhibitor Assay

The assay of clathrin-mediated inhibitor (chlorpromazine, CP), which was purchased from MedChemExpress (Shanghai, China), was performed as described in a previous study [20]. Specifically, the differentiated Caco-2 cells were washed with Hank’s balanced salt solution (HBSS), and then the DMEM cell culture medium was replaced with HBSS or HBSS containing CP 60 μM [20]. After a 60 min treatment, the TEER value was measured, and a confocal study was conducted as described above.

### 4.16. Statistical Analyses

The GraphPad Prism 5.0 (San Diego, CA, USA) was used to analyze the experimental data and carry out the statistical analysis through a one-way analysis of variance (ANOVA) followed by Tukey’s multiple comparison tests. Data were expressed as the mean ± standard deviation (SD) of 3 independent experiments with 9 replicates. A *p*-value of 0.05 was considered statistically significant.

## Figures and Tables

**Figure 1 toxins-13-00184-f001:**
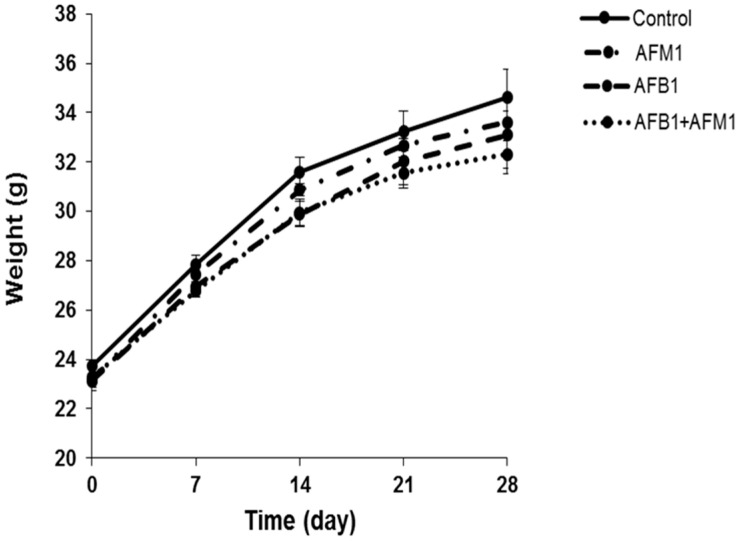
Individual and combined effects of aflatoxin B1 (AFB1) and aflatoxin M1 (AFM1) on body weight of mice. Male Institute of Cancer Research (ICR) mice were divided into four groups: control (1% dimethylsulfoxide, DMSO), exposed to individual AFB1 (0.3 mg/kg b.w.), individual AFM1 (3.0 mg/kg b.w.), and combined AFB1 and AFM1 (0.3 + 3.0 mg/kg b.w.). Values represent mean ± SD (*n* = 10 animals).

**Figure 2 toxins-13-00184-f002:**
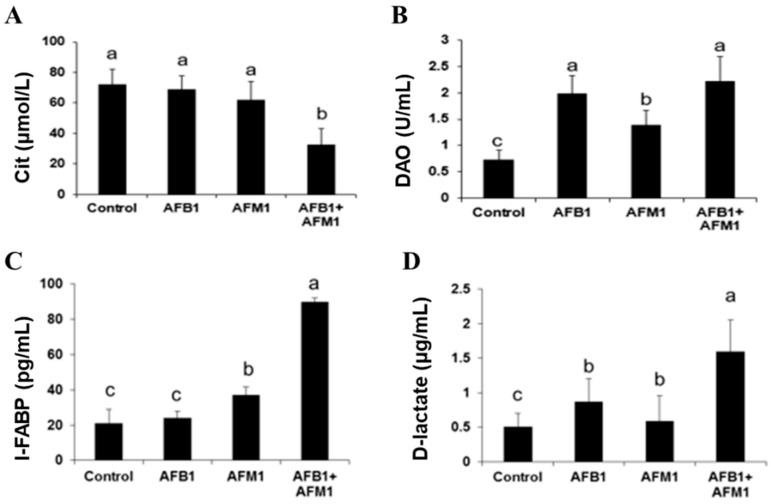
Individual and combined effects of AFB1 and AFM1 on serum citrulline (Cit) (**A**), diamine oxidase (DAO) (**B**), intestinal fatty acid-binding protein (I-FABP) (**C**), and D-lactate (**D**) levels in ICR mice. Values represent mean ± SD (*n* = 10 animals). Means with a different letter (a, b and c) differ significantly (*p* < 0.05).

**Figure 3 toxins-13-00184-f003:**
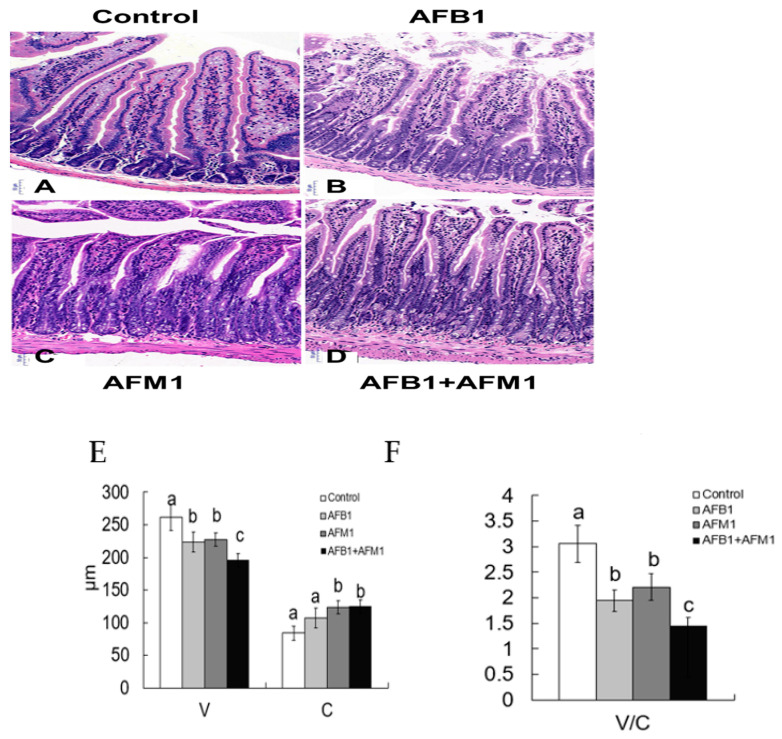
Individual and combined effects of AFB1 and AFM1 on ileum histology. Histology of the ileum after hematoxylin-eosin staining (HE, 10×): control mice (1%DMSO, (**A**)), AFB1 treated mice (**B**), AFM1 treated mice (**C**), and AFB1+AFM1 treated mice (**D**), villus height and crypt depth (**E**), and the ratio of villus height/crypt depth (**F**). Values represent mean ± SD (*n* = 6 animals). Means with a different letter (a, b and c) differ significantly (*p* < 0.05).

**Figure 4 toxins-13-00184-f004:**
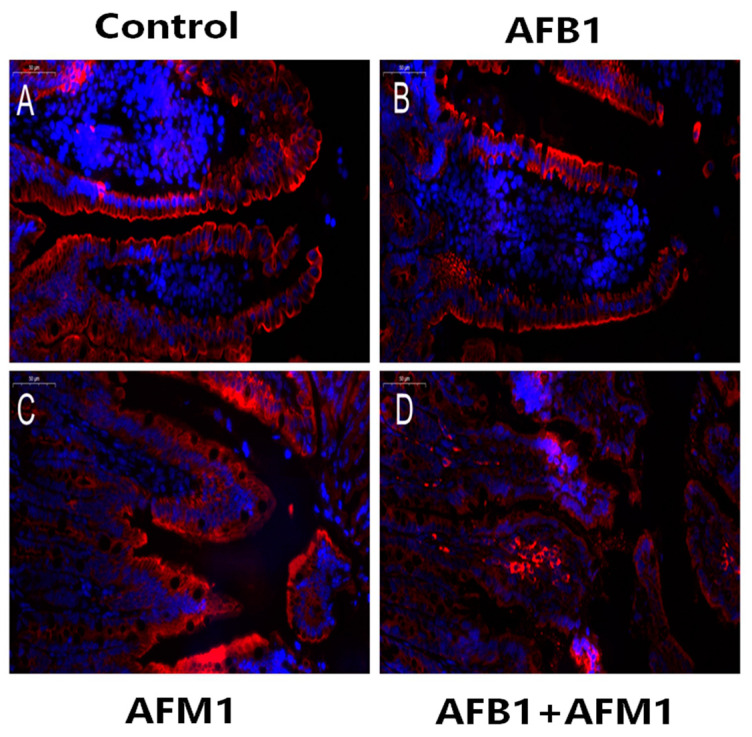
Individual and combined effects of AFB1 and AFM1 on claudin-1 expression and distribution in the ileum measured by immunofluorescence staining (400×). The ileum of ICR mice in the control group (1% DMSO, (**A**)) exerted strong homogeneous staining on claudin-1 with a distinct latticed structure. Less intense immunostaining of claudin-1 with an unclear latticed structure occurred in AFB1-treated mice (**B**), AFM1-treated mice (**C**), and AFB1+AFM1-treated mice (**D**). *n* = 6 animals.

**Figure 5 toxins-13-00184-f005:**
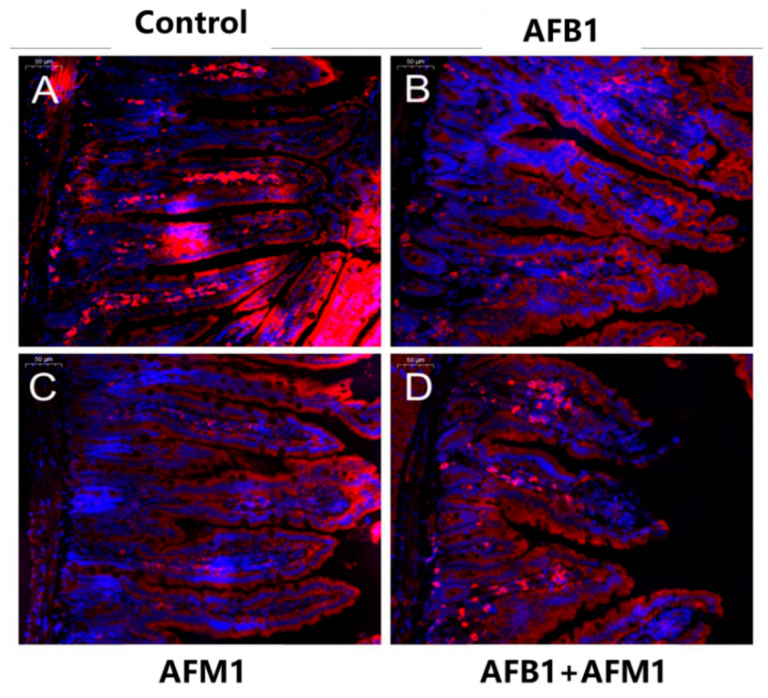
Individual and combined effects of AFB1 and AFM1 on ZO-1 expression and distribution in ileum measured by immunofluorescence staining (200×). Ileum of ICR mice in the control group (1% DMSO, (**A**)) exerted strong homogeneous staining on ZO-1 with a distinct latticed structure. Less intense immunostaining of ZO-1 with an unclear latticed structure occurred in AFB1-treated mice (**B**), AFM1-treated mice (**C**), and AFB1+AFM1-treated mice (**D**). *n* = 6 animals.

**Figure 6 toxins-13-00184-f006:**
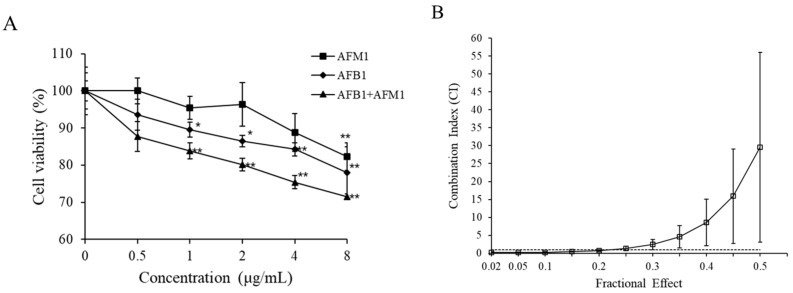
Cytotoxic effects of AFB1 and AFM1 on differentiated Caco-2 cells. The differentiated Caco-2 cells were exposed to individual and combined AFB1 and AFM1 for 48 h. Viability of Caco-2 cells was detected by CCK-8 kit (**A**). CI/fa curve based on isobologram analysis using CalcuSyn software (**B**). Data represent the mean of three independent experiments ± SD (*n* = 4). * *p* < 0.05, ** *p* < 0.01 represent significance, compared with the control group.

**Figure 7 toxins-13-00184-f007:**
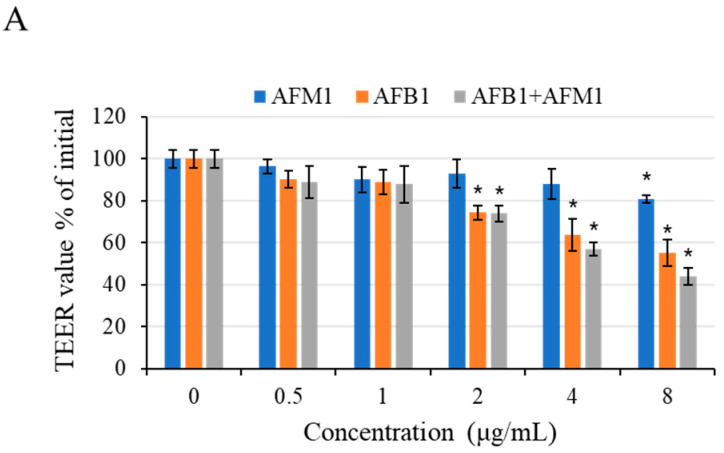
Intestinal permeability effects of AFB1 and AFM1 on differentiated Caco-2 cells. Cells were cultured with individual and combined AFB1 and AFM1 at concentrations of 0.5, 1, 2, 4, and 8 μg/mL for 48 h. Trans-epithelial electrical resistance (TEER) assay (**A**) and fluorescein isothiocyanate (FITC) paracellular permeability (**B**), (**C**) were determined. Data represent the mean of three independent experiments ± SD (*n* = 4). * *p* < 0.05, ** *p* < 0.01, *** *p* < 0.001 represent significance compared with the control group.

**Figure 8 toxins-13-00184-f008:**
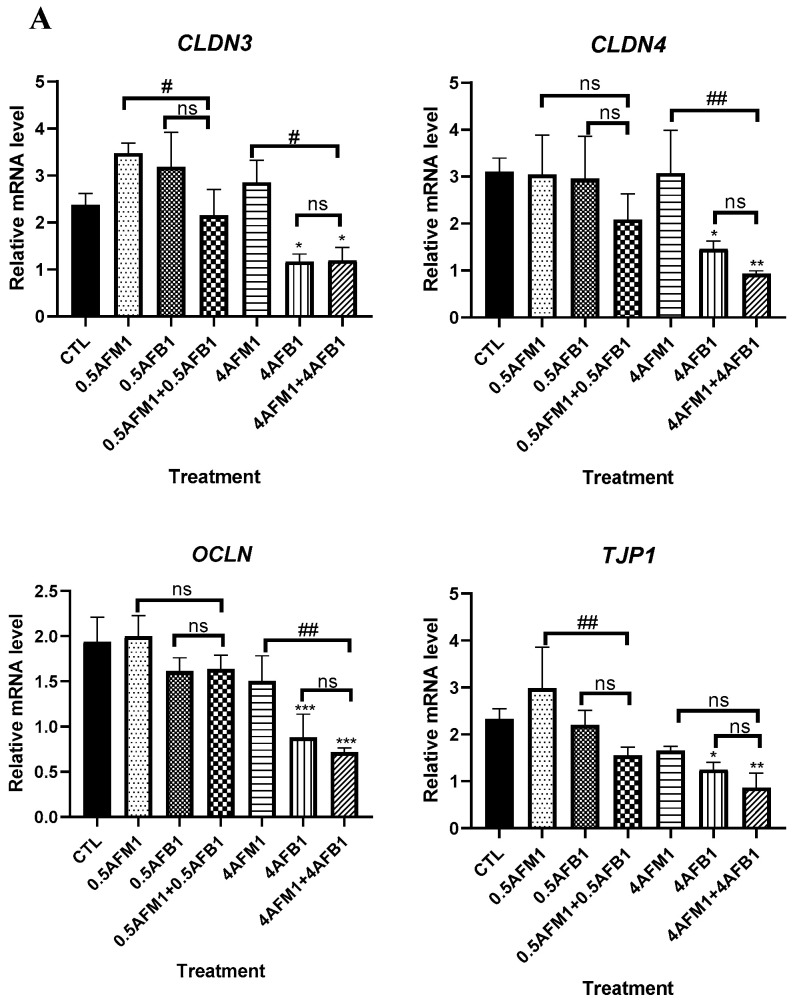
Effects of non-toxic AFM1 on AFB1-induced tight junction (TJ) destruction on differentiated Caco-2 cells. Cells were treated with AFB1 (0.5 and 4 μg/mL) and AFM1 (0.5 and 4 μg/mL) individually and collectively for 48 h. mRNA expression measured by quantitative real-time PCR (qRT-PCR) assay (**A**) and protein expression measured by western blot assay ((**B**,**C**)). The distribution of TJ proteins on differentiated Caco-2 cells was visualized (**D**). Reference gene *GAPDH* and protein β-actin were used as a control for qRT-PCR and western blot assay, respectively. The images shown are representative of at least 3 regions observed on the same slide, and 3 different sections were analyzed for each condition. Data represent the mean of three independent experiments ± SD (*n* = 4). * *p* < 0.05, ** *p* < 0.01, *** *p* < 0.001 represent significance compared to the control group. # *p* < 0.05, ## *p* < 0.01 represent significance compared with individual AFB1 or AFM1 treatments. ns represents *p* > 0.05 between two treatments.

**Figure 9 toxins-13-00184-f009:**
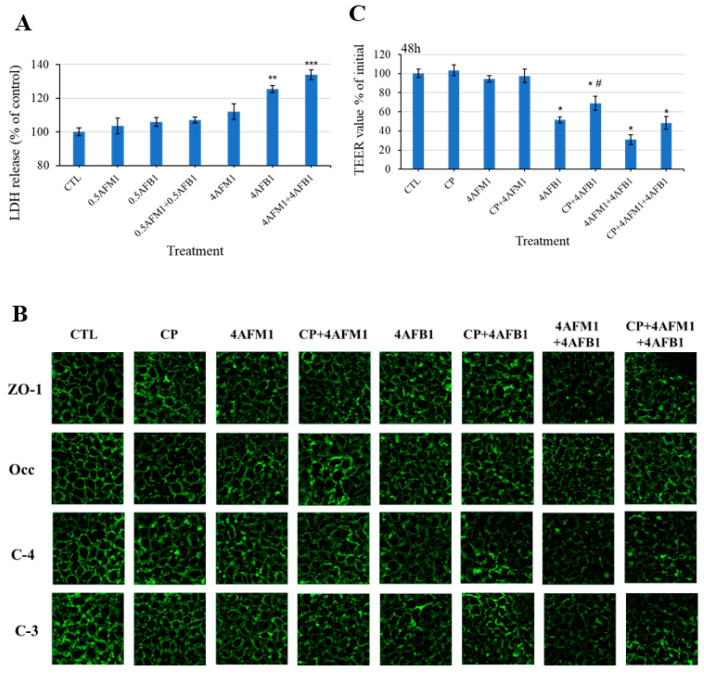
Potential mechanisms of the internalization of TJ proteins caused by AFB1 and AFM1 in differentiated Caco-2 cells. Lactate dehydrogenase (LDH) leakage (**A**) from cells into the culture medium was measured to evaluate the damage of the plasma membrane. TEER values (**B**) were measured, and the distribution of TJ proteins (**C**) was assessed after preincubation with 60 μM CP for 60 min, followed by toxins treatment for 48 h. Data represent the mean of three independent experiments ± SD. * *p* < 0.05, ** *p* < 0.01, *** *p* < 0.001 represent significance compared with the control group. # *p* < 0.05 represents significance compared with individual AFB1 or AFM1 treatments.

**Figure 10 toxins-13-00184-f010:**
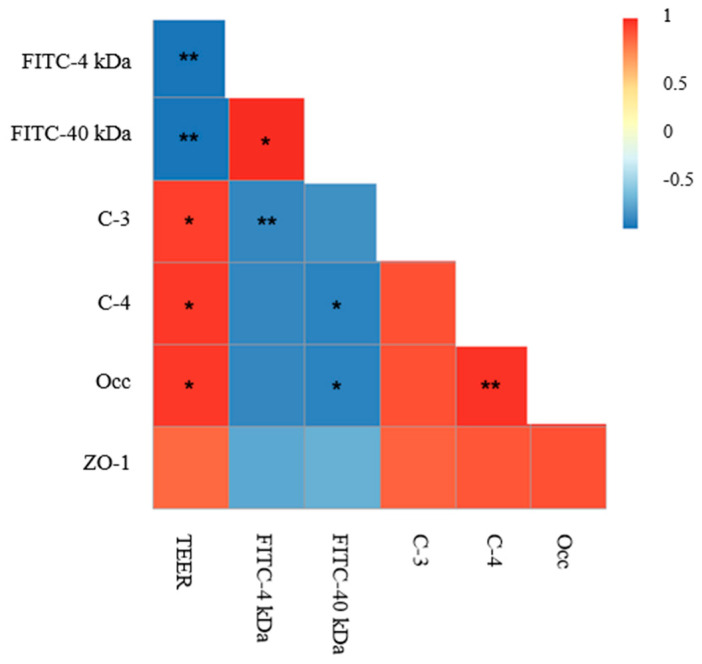
Correlation analysis for TEER values, FITC-dextran, and expression of TJ proteins. A heat map was generated with the measured data and was shown as a ladder diagram. As the color key shows, the colors blue, yellow, and red represent a negative correlation, a low correlation, and a positive correlation, respectively. Spearman’s correlations were used to analyze the statistical significance. * *p* < 0.05, ** *p* < 0.01.

**Figure 11 toxins-13-00184-f011:**
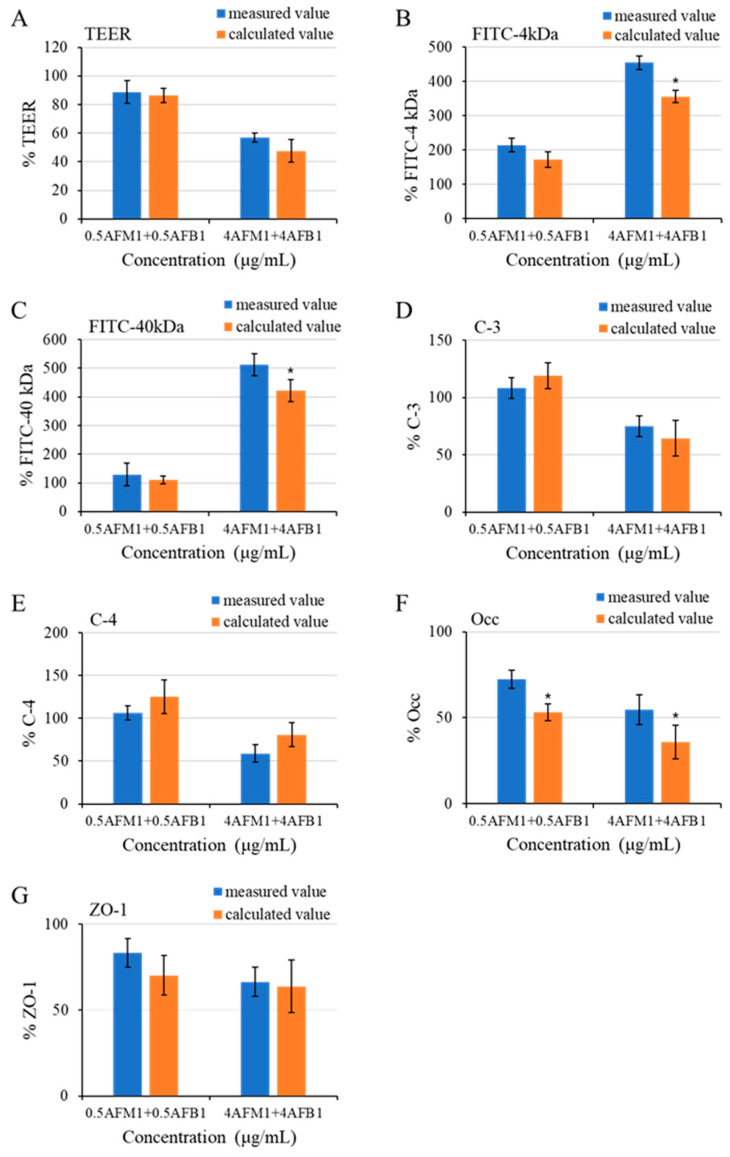
Interactive effects of combinations of AFB1 and AFM1 on differentiated Caco-2 cells on various measured endpoints (**A**–**G**). The calculated value represents expected values, and the calculation method is shown in the material method section. Data for measured parameters are expressed in % relative to the control group. * *p* < 0.05 representing either significant synergistic or antagonist effect.

**Table 1 toxins-13-00184-t001:** Primers of tight junction (TJ) proteins.

Gene	Primers
*OCLN*	Forward: 5′- CTCTCAGCCAGCCTACTCTT-3′ Reverse: 5′- TAGCCATAGCCATAGCCACTT-3′
*CLDN3*	Forward: 5′- CCTTCATCGGCAGCAACATC-3′ Reverse: 5′- GCAGCGAGTCGTACACCTT-3′
*CLDN4*	Forward: 5′- CGTCATCATCAGCATCATCGT-3′ Reverse: 5′- CACCAGCGGATTGTAGAAGTC-3′
*TJP1*	Forward: 5′- GCTGTGGAAGAGGATGAAGATG-3′ Reverse: 5′- AGGTGGAAGGATGCTGTTGT-3′
*GAPDH*	Forward: 5′- GAAGGTGAAGGTCGGAGTC-3′ Reverse: 5′- GAAGATGGTGATGGGATTTC-3′

## Data Availability

All the data generated for this study are included in the article.

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
