# Peer review of "Aflatoxin B1 and Aflatoxin M1 Induce Compromised Intestinal Integrity through Clathrin-Mediated Endocytosis"

_toxins, 2021, doi:10.3390/toxins13030184_

Round 1
Reviewer 1 Report
Section 2.2: concerning the dose of AFB1 and AFM1 the Authors refer to the LD50 values. It should be more corrected to relate the doses to the possible chronic exposure related to their intake.
It is not fully clear the choice of doses for the in vivo experiment for both AFB1 and AFM1, and mainly for the doses of aflatoxin M1. The choice of concentrations for the in vitro experiment should also be well supported by an explanation and this is another critical aspect that the Authors are asked to clarify.
Section 2.6: the citotoxic assay should be better described (e.g. cell procedures), the used concentrations should also be discussed.
Section 2.11: for antibodies it should be indicated whether purchased or homemade. If purchased, give indications for the repeatability of the experiment.
Figure 3: the picture should be rotated with the intestinal villus vertically.
Figure 3, 4 and 5: in all three cases it is indicated that the number of specimens used is 6. Why not 10 (or 11) as done for other essays?
Figure 8 B: the molecular weight expressed in kD must be entered next to the picture.
Author Response
Dear reviewer,
Thank you very much for providing an opportunity for us to revise our paper, entitled ‘Aflatoxin B1 and Aflatoxin M1 Induce Compromised Intestinal Integrity Through Clathrin-Mediated Endocytosis’ (Manuscript Toxins 1095036) for your journal. And thank you so much for the constructive comments and valuable suggestions on the manuscript.
According to the valuable advices, we have made a minor revision to this manuscript. The manuscript has been checked by an English native speaker to avoid the grammar error.
Responds to reviewers' comments:
Reviewer #1:
(1): Section 2.2: concerning the dose of AFB1 and AFM1 the Authors refer to the LD50 values. It should be more corrected to relate the doses to the possible chronic exposure related to their intake.
AU: Thanks for your suggestions. As you suggested, it should be more corrected to relate the doses to the possible chronic exposure related to intake. We added the related description as ‘The doses used in the present study are several orders of magnitude higher than toxin levels to the human possible chronic exposure intake.’ in line 104-105 of the revised manuscript.
In addition, we also have discussed the related contents as ‘A critical reflection
about the doses chosen in the in vivo part of the present study and concentrations applied to Caco-2 cells is indicated: these are several orders of magnitude higher than toxin levels found in the serum of human beings’ in line 449-451 of the revised manuscript. And the detailed mycotoxins’ doses in human under chronic exposure to contaminated food and environment were shown in line 452-467 of the revised manuscript.
(2): It is not fully clear the choice of doses for the in vivo experiment for both AFB1 and AFM1, and mainly for the doses of aflatoxin M1. The choice of concentrations for the in vitro experiment should also be well supported by an explanation and this is another critical aspect that the Authors are asked to clarify.
AU: Thanks for your suggestions. In the present study, due to the price of AFM1 is much higher than AFB1, we only performed the preliminary test of AFB1 to obtain its LD50 values. And then we calculated the LD50 values of AFM1. The doses of AFB1 and AFM1 used in the present study were 1/20 of their LD50 values. We clarified the doses of AFM1 used in the present study as ‘AFM1 was proved to induce liver cancer with a potency one-tenth that of AFB1. And in general, for AFM1, a potency factor of 0.1 relative to AFB1 was used [23], so we estimated the LD50 values for AFM1 were 60 mg/kg b.w. Therefore, dose of 3.0 mg/kg b.w. AFM1 (1/20 of the LD50 values) was chosen in this study.’ in line 100-104 of the revised manuscript.
In addition, as your suggestions, we added the reasons on the choice of
concentrations for AFB1 and AFM1 in line 170-173 of the revised manuscript as
‘Based on the results of cell viability and epithelial integrity (TEER and FITCdextran), AFM1 (0.5 and 4 μg/mL) and corresponding concentrations of
AFB1were selected in the subsequent experiments to determine the effects of noncytotoxic AFM1 on AFB1’.
(3): Section 2.6: the citotoxic assay should be better described (e.g. cell procedures), the used concentrations should also be discussed.
AU: Thanks for your suggestions. Firstly, as your suggestions, we added the details of cytotoxicity assay as ‘The cytotoxicity of the compounds was analyzed using the CCK-8 assay (Beyotime, Shanghai, China) according to the manufacturer’s instructions. In brief, individual and combined AFB1 and AFM1 (0.5, 1, 2, 4 and 8 μg/mL) were added to the apical and basal compartments of transwell chambers. After 48 h, 120 μL CCK-8 operating reagent was added to the apical side of every well in permeable Millicell PCF filters. After about 1 h incubation at 37℃, we pipetted out 100 μL liquor to measure the absorbance at a test wavelength of 450 nm using an automated ELISA reader (Thermo Scientific, Waltham, MA, USA)’ in line 148-154 of the revised manuscript.
In addition, we have discussed the used concentrations as ‘Based on the results of cell viability and epithelial integrity (TEER and FITC-dextran), AFM1 (0.5 and 4
μg/mL) and corresponding concentrations of AFB1were selected in the
subsequent experiments to determine the effects of non-cytotoxic AFM1 on
AFB1.’ in line 170-173 of the revised manuscript.
(4): Section 2.11: for antibodies it should be indicated whether purchased or homemade. If purchased, give indications for the repeatability of the experiment.
AU: Thanks for your kindly suggestions. The antibodies used in the present study were purchased from Bioss (Beijing, China), which described in line 89-92 of the
revised manuscript. And as your suggestions, we have modified the related
description as ‘The membranes were incubated with primary antibodies of 4
different TJ proteins (as mentioned above in section 2.1) overnight at 4℃’ in line
of 202-203 the revised manuscript. And we also added the repeatability of the
experiment as ‘The results of western blotting are expressed as mean ± SD of three independent experiments’ in line 207-208 of the revised manuscript.
(5): Figure 3: the picture should be rotated with the intestinal villus vertically.
AU: Thanks for your kindly suggestions. As your suggestions, we have modified Figure 3A-D with the intestinal villus vertically in the revised manuscript.
(6): Figure 3, 4 and 5: in all three cases it is indicated that the number of specimens used is 6. Why not 10 (or 11) as done for other essays?
AU: Thanks for your kindly question. Firstly, total 10 four-week old male ICR mice in each treatment were used in the present study. Apart from the histological and morphometric assessment as well as immunofluorescence analysis of the ileum in mice, we also performed the proteome assay, which is not shown in the present study. Therefore, to ensure the amount of sample required for proteomic research, the number of specimens used in the phenotypic experiments is 6.
Secondly, as previous studies showed that it is appropriate to set 5-6 replicates per treatment in animal experiments (Chandratre et al., 2014; Guan et al., 2015; Zhao et al., 2016), therefore, in the present study, we used 6 mice in each treatment.
Chandratre, G.A.; Telang, A.G.; Badgujar, P.C.; Raut, S.S.; Sharma, A.K.
Toxicopathological alterations induced by high dose dietary T-2 mycotoxin and its residue detection in Wistar rats. Arch Environ Contam Toxicol 2014, 67, 124-138, doi:10.1007/s00244-014-0006-x.
Guan, L.Z.; Sun, Y.P.; Cai, J.S.; Wu, H.D.; Yu, L.Z.; Zhang, Y.L.; Xi, Q.Y. The
aflatoxin-detoxifizyme specific expression in mouse parotid gland. Transgenic Res 2015, 24, 489-496, doi:10.1007/s11248-015-9863-y.
Zhao, L.; Li, X.; Ji, C.; Rong, X.; Liu, S.; Zhang, J.; Ma, Q. Protective effect of
Devosia sp. ANSB714 on growth performance, serum chemistry, immunity
function and residues in kidneys of mice exposed to deoxynivalenol. Food Chem
Toxicol 2016, 92, 143-149, doi:10.1016/j.fct.2016.03.020.
(7): Figure 8 B: the molecular weight expressed in kD must be entered next to the picture.
AU: Thanks for your kindly suggestions. As your suggestions, we have modified Figure 8B and Figure 8C in which the molecular weight expressed in kD was entered in the Y-axis in the revised manuscript.
Reviewer 2 Report
The authors demonstrate in vivo effects of individual and combined use of toxins AFB1 and AFM1 by applying a number of read-outs. These analysis indicated that the toxins might affect intestinal barrier integrity. Next, they applied Caco-2 to confirm barrier impeding effects of toxins and subsequently identify the mechanistic basis for these effect. The applied methods are cohesive and a logical order which provides a clear insight. The novelty of this work is, however, low as a similar combination of experiments, minus the animal trial, was applied previously (REF 22) to determine the impact of AFM1 and/or OTA toxin on Caco-2 and the most active toxin (AFB1) has been investigated in many previous studies with similar assays. The combination of these toxins, however, is rather novel providing a new message in this manuscript and potential implications.
Major comments:
Authors applied immunohistochemical analysis to identify effects on TJ proteins in vivo. In general, the presented findings were overinterpreted and require software based analysis to substantiate the conclusions.
lines 271/272: I assume the pictures in figure 4 are representative of the n=6? The conclusion on, in particular, AFB1, needs to be substantiated with objective measurements as the difference with control is minimal at best. The dispersion of claudin intracellularly in the AFM1 group is more clear, but software based analysis to objectify the conclusion is preferred.
line 276: from these images individual cells are difficult to distinguish, and therefore cytoplasmic vs membrane location is also difficult to conclude on. higher resolution images would be needed for these conclusions
lines 279-281: the conclusions on ZO-1 are not supported by the presented data. the re-distribution of ZO-1 to the basolateral membrane is not evident from these pictures.
389-391: similar overinterpretation of confocal images
Throughout the description and analysis of the Caco-2 results, the authors overinterpret data to support a conclusion that the combination of the toxins is more effective than AFM1 alone. These needs to adapted.
lines 317/318: the word drastic cannot be applied when there is no significant difference between the combination and AFB1 alone. Moreover, even suggesting that the combination has a more detrimental effect, being a trend, than AFB1 alone based on these data is unwarranted.
line 324: this conclusion is not supported by the data. indeed, the combination allows increased passage consistently over the various concentrations, but significance analysis does not indicate an added effect of AFB1
line 340/341: again not support by the data. the only observed significance results from increased TJ protein expression by AFM1 when compared to the control (based on AVG). of unrelated note, I am surprised that the increase in CLD3 expression by AFM1 when compared to control did not reach significance. this is in line with described findings in the discussion on lines 506-507. might this be related to continuous, rather than accute, exposure to the toxins?
line 344-346: conclusion not supported by data
line 352-353: in contrast to all other claims above, the data here actually would support this conclusion as significance is only reached for the combination for CLDN3, CLD4 and OCCL and ZO-1 at the lower concentration
line 360-366: not supported by objective measurements so not a conclusion that can be drawn here. The images
Based on the above comments the authors would need to significantly re-evaluate the discussion for the reporting of their findings and described implications.
Minor comment:
line 46: is AFB1 converted into AFM1 in the organism? this is not evident from the text and not clear to reader's unfamiliar with these toxins. How does this impact the toxicity of the toxin as AFB1 is shown to be the more active compound? This latter might be relevant for the discussion
figure 8D: this data was acquired by growing Caco-2 on cover slips. Can the authors confirm, or indicated references, that this growth method, similar to transwells, induces polarisation and differentiation of the cells? If not, authors should indicate that these findings were acquired with a different caco-2 culturing method. In general, the findings still support the analysis and conclusions as authors investigate total expression pattern and not specifically apical or basolateral localization of TJ proteins.
lines 460-474: here the authors explain how minute levels of toxins can be found in the sera of people. Given the fact that the mice were exposed to higher levels, detection of toxin in the sera of mice is to be expected, and would substantiate the importance of this study as is links to this section in the discussion.
Author Response
Dear reviewer,
Thank you very much for providing an opportunity for us to revise our paper, entitled “Aflatoxin B1 and Aflatoxin M1 Induce Compromised Intestinal Integrity Through Clathrin-Mediated Endocytosis” (Manuscript Toxins 1095036) for your journal. And thank you so much for the constructive comments and valuable suggestions on the manuscript.
According to the valuable advices, we have made a minor revision to this manuscript. The manuscript has been checked by an English native speaker to avoid the grammar error.
Reviewer #2:
Authors applied immunohistochemical analysis to identify effects on TJ proteins in vivo. In general, the presented findings were overinterpreted and require software based analysis to substantiate the conclusions.
(1): lines 271/272: I assume the pictures in figure 4 are representative of the n=6? The conclusion on, in particular, AFB1, needs to be substantiated with objective measurements as the difference with control is minimal at best. The dispersion of claudin intracellularly in the AFM1 group is more clear, but software based analysis to objectify the conclusion is preferred.
AU: Thanks for your kindly suggestions. Firstly, as shown in the legends in line 290-291 and 297, the pictures in figure 4 and figure 5 are representative of the n=6 animals.
Secondly, as your suggestions, to describe the results of immunofluorescence
staining more objectively, we deleted the related description in original
manuscript, and streamlined the contents in line 263-268 of the revised manuscript as ‘To study the effects of AFB1 and AFM1 alone and in combination on the TJ structure of mice intestinal epithelial cells, the abundance of claudin-1 and ZO-1 were assessed in the ileum by immunohistochemical staining. Form the results, we observed that a weak immunostaining intensity and a clear collapse of the gridlike barrier structure were shown in the mice exposed to the mixture (Fig. 4, 5).’
(2): line 276: from these images individual cells are difficult to distinguish, and
therefore cytoplasmic vs membrane location is also difficult to conclude on. higher resolution images would be needed for these conclusions
AU: Thanks for your kindly suggestions. As the answer to question 1, to describe the results of immunofluorescence staining more objectively, we deleted the related description in original manuscript, and streamlined the contents in line 263-268 of the revised manuscript as ‘To study the effects of AFB1 and AFM1 alone and in combination on the TJ structure of mice intestinal epithelial cells, the abundance of claudin-1 and ZO-1 were assessed in the ileum by
immunohistochemical staining. Form the results, we observed that a weak
immunostaining intensity and a clear collapse of the grid-like barrier structure
were shown in the mice exposed to the mixture (Fig. 4, 5).’
(3): lines 279-281: the conclusions on ZO-1 are not supported by the presented data. the re-distribution of ZO-1 to the basolateral membrane is not evident from these pictures.
AU: Thanks for your kindly suggestions. As the answer to question 1 and question 2, to describe the results of immunofluorescence staining more objectively, we deleted the related description in original manuscript, and streamlined the contents in line 263-268 of the revised manuscript as ‘To study the effects of AFB1 and AFM1 alone and in combination on the TJ structure of mice intestinal epithelial cells, the abundance of claudin-1 and ZO-1 were assessed in the ileum by immunohistochemical staining. Form the results, we observed that a weak immunostaining intensity and a clear collapse of the grid-like barrier structure were shown in the mice exposed to the mixture (Fig. 4, 5).’
Secondly, we deleted the related discussion about immunofluorescent staining
results of original manuscript line 502-505 and 507-510 in the revised manuscript.
(4): 389-391: similar overinterpretation of confocal images
AU: Thanks for your kindly suggestions. As your suggestions, to describe the results of immunofluorescence staining more objectively, we modified the related description as ‘Fig. 9B shows that compared with the combined mycotoxins treatment, the immuno-fluorescence intensity was stronger in the presence of preincubation with CP, especially for claudin-4.’ in the line 377-379 of the revised manuscript.
Throughout the description and analysis of the Caco-2 results, the authors overinterpret data to support a conclusion that the combination of the toxins is more effective than AFM1 alone. These needs to adapted.
(5): lines 317/318: the word drastic cannot be applied when there is no significant difference between the combination and AFB1 alone. Moreover, even suggesting that the combination has a more detrimental effect, being a trend, than AFB1 alone based on these data is unwarranted.
AU: Thanks for your kindly suggestions. As your suggestions, to describe relevant results more accurately, we deleted the related contents in the revised manuscript.
(6): line 324: this conclusion is not supported by the data. indeed, the combination allows increased passage consistently over the various concentrations, but significance analysis does not indicate an added effect of AFB1
AU: Thanks for your kindly suggestions. As your suggestions, to describe relevant results more accurately, we deleted the related contents in the revised manuscript.
(7): line 340/341: again not support by the data. the only observed significance results from increased TJ protein expression by AFM1 when compared to the control (based on AVG). of unrelated note, I am surprised that the increase in CLD3 expression by AFM1 when compared to control did not reach significance. this is in line with described findings in the discussion on lines 506-507. might this be related to continuous, rather than accute, exposure to the toxins?
AU: Thanks for your kindly suggestions. Firstly, as your suggestions, we deleted the related contents in the revised manuscript.
Secondly, we checked the data again and found that the p value between 0.5
μg/mL AFM1 and control treatment in CLD3 transcript expression was 0.07, did
not reach significance.
Thirdly, we modified the related described findings in the discussion in line 525-
529 as ‘In the present study, 0.5 μg/mL AFM1 tends to increase the transcription
expression of claudin-3. This is consistent with a study showed that an increase in transcript levels of claudin-1 and claudin-2 in the jejunum of broiler chicks fed
with 1.5 mg/kg AFB1 for 20 days was observed [52], which might be related to
continuous rather than acute exposure to toxins.’ in the revised manuscript.
(8): line 344-346: conclusion not supported by data
AU: Thanks for your kindly suggestions. As your suggestions, we deleted the related description in the revised manuscript.
(9): line 352-353: in contrast to all other claims above, the data here actually would support this conclusion as significance is only reached for the combination for CLDN3, CLD4 and OCCL and ZO-1 at the lower concentration
AU: Thanks for your kindly suggestions. As your suggestions, to describe the related results more accurately, we modified the description as ‘Compared with control group, the mixture of AFB1 and AFM1 at higher concentration produced more serious damage of TJ proteins than individual toxins.’ in line 346-347 of the revised manuscript.
(10): line 360-366: not supported by objective measurements so not a conclusion that can be drawn here. The images
AU: Thanks for your kindly suggestions. As your suggestions, we deleted the related description in the revised manuscript.
(11): Based on the above comments the authors would need to significantly re-evaluate the discussion for the reporting of their findings and described implications.
AU: Thanks for your kindly suggestions. As your suggestions, modified the discussion in the revised manuscript.
Firstly, we modified the related description as ‘We found that AFB1 and AFM1
induced intestinal integrity dysfunction in differentiated Caco-2 cells through
clathrin-mediated endocytosis on TJ proteins’ in line 432-434 of the revised
manuscript.
Secondly, we deleted the description of ‘Considering also the individual and
combined toxin results, we concluded that AFM1 aggravated AFB1-induced
compromised intestinal integrity.’ in the discussion of the revised manuscript.
Thirdly, we modified the related description as ‘From the above results of TEER
values, FITC-dextran and TJ proteins, we confirmed that AFB1 and AFM1
compromised intestinal barrier function’ in line 542-543 of the revised manuscript.
(12): line 46: is AFB1 converted into AFM1 in the organism? this is not evident from the text and not clear to reader's unfamiliar with these toxins. How does this impact the toxicity of the toxin as AFB1 is shown to be the more active compound? This latter might be relevant for the discussion
AU: Thanks for your kindly suggestions. Firstly, as your suggestions, we added more detailed description on AFB1 converted into AFM1 in the organism as ‘The
metabolism and activation of AFB1 can take place in the liver by cytochrome
P450 (CYP) enzymes (mainly by CYP1A2 and CYP3A4) [11]. While it has been
largely described in liver, AFB1 metabolism to its toxic metabolite also occurs
in the digestive tract [12]. Apart from the epoxidation, it can also be oxidized by
P450-dependent monooxygenases to hydroxylated products, such as aflatoxin
M1 (AFM1), which is the most threatening component of AFB1 contamination
[13].’ in line 46-51 of the revised manuscript.
Secondly, as your suggestions, we added the related description as ‘This may be
related to the fact that AFB1 is shown to be the more active compound, which is
more potent than AFM1 with respect to liver carcinogenicity by approximately
10-fold [23]’ in line 498-500 of the revised manuscript.
(13): figure 8D: this data was acquired by growing Caco-2 on cover slips. Can the
authors confirm, or indicated references, that this growth method, similar to
transwells, induces polarisation and differentiation of the cells? If not, authors
should indicate that these findings were acquired with a different caco-2 culturing method. In general, the findings still support the analysis and conclusions as authors investigate total expression pattern and not specifically apical or basolateral localization of TJ proteins.
AU: Thanks for your kindly suggestions. As your suggestions, we added the indicated references as ‘Caco-2 cells (2 × 105 per well) were cultured on coverslips in 6 well plates for 21 days [24,25]’ in line 178 of the revised manuscript.
(14): lines 460-474: here the authors explain how minute levels of toxins can be found in the sera of people. Given the fact that the mice were exposed to higher levels, detection of toxin in the sera of mice is to be expected, and would substantiate the importance of this study as is links to this section in the discussion.
AU: Thanks for your suggestions. As your suggestions, we have added the related contents as ‘A previous study demonstrated that AFB1 residues in blood of the AFB1-fed treatment were significantly higher (p<0.05) than those of the fed with standard mouse feed for 14 days [37]. Similar results also showed in the
Juvenile Turbot fed with AFB1 diets [38]. Not only in the blood, mycotoxins
could also residue in the various tissues in mice and rats, such as liver and kidney [39-41].’ in line of the revised manuscript.